# Microsaccades strongly modulate but do not directly cause the EEG N2pc marker of spatial attention

**Baiwei Liu***, **Siyang Kong, Freek van Ede***

Department of Experimental and Applied Psychology, Institute for Brain and Behavior Amsterdam, Vrije Universiteit Amsterdam, Amsterdam, The Netherlands

* b.liu@vu.nl (BL); freek.van.ede@vu.nl (FvE)

## Abstract

The N2pc is a popular human-neuroscience marker of covert and internal spatial attention that occurs 200–300 ms after being prompted to shift attention—a time window also characterized by the spatial biasing of small fixational eye movements known as microsaccades. Here, we show how co-occurring microsaccades profoundly modulate N2pc amplitude during top-down shifts of spatial attention in both perception and working memory. At the same time, we show that a significant—albeit severely weakened—N2pc can still be established in the absence of co-occurring microsaccades. Moreover, despite the strong modulation of the N2pc by microsaccade presence and direction, the N2pc does not align to the precise timing of microsaccades, ruling out that the observed N2pc modulations by microsaccades are a direct artifact of microsaccade-related eye-muscle activity, corneo-retinal dipole movement, or visual inputs moving over the retina. Thus, while microsaccades strongly modulate N2pc amplitude, microsaccades themselves are not a prerequisite for, nor a direct cause of, the N2pc.

## Introduction

Attention is the foundational process by which we selectively process and prioritize information that is relevant to us [1,2], a central topic within the study of mind and brain. Attention can be deployed covertly to locations outside of current fixation, as well as internally to representations held in working memory [3,4]. A widely used marker of covert and internal attention shifts in human neuroscience is the N2pc—a lateralized posterior EEG potential that emerges around 200 ms after being prompted to shift attention. Since the discovery of the N2pc around 30 years ago [5], this marker has provided invaluable insights into the principles, mechanisms, and timings of covert and internal attention shifts [5–15].

A vital assumption in laboratory studies of the N2pc is that participants remain their current fixation while shifting attention. Only then can the spatial modulation of

**Data availability statement:** All analysis code, including the scripts and the data directly used to generate the figures, are available on OSF (https://osf.io/v3ct7/). The raw EEG, eye-tracking, and behavioral data can be found in Zenodo (https://zenodo.org/records/17106015).

**Funding:** This work was supported by an ERC Starting Grant from the European Research Council (MEMTICIPATION, 850636) and an NWO Vidi Grant from the Dutch Research Council (14721) to F.v.E.. The funders had no role in study design, data collection and analysis, decision to publish, or preparation of the manuscript.

**Competing interests:** The authors have declared that no competing interests exist.

**Abbreviations:** ERP, event-related potential; ICA, Independent Component Analysis; NaN, Not-a-Number.

brain activity, like the N2pc, be attributed to a top-down cognitive modulation associated with attention (as opposed to consequences of eye movements or the retinal displacements they trigger). The validity of this vital assumption, however, is not guaranteed. Even when instructed to keep fixation, humans nevertheless produce small—often overlooked—eye movements known as microsaccades [16–18]. Critically, the direction of microsaccades has also been shown to be modulated by the top-down deployment of covert [19–21] and internal [21–26] spatial attention—with profound influences on neuronal processing even for peripheral visual inputs [27–31]. Moreover, spatial modulations in microsaccades during covert and internal attention shifts occur at time windows that overlap with the N2pc, also emerging around 200 ms after being prompted to shift attention. This raises the critical question to what extent the N2pc is contingent on—or even caused by—co-occurring spatial biases in microsaccades.

Despite the extensive use of the N2pc for studying top-down attention, the potential contribution of microsaccades to the N2pc remains elusive (unlike for complementary neural attention markers [22,27,32,33] or spatial event-related potential (ERP) modulations associated with bottom-up cue processing [34]). Moreover, prior studies on the link between microsaccades and attention almost exclusively considered covert shifts of spatial attention in the context of perception. Here, microsaccades may influence neural activity through retinal displacement of the attended sensory inputs. In contrast, when shifting attention to visual contents held in visual working memory, the potential contribution of physical displacement of the attended object over the retina is sidestepped because the attended object is in memory. To address these outstanding points, we investigated the contribution of microsaccades to the N2pc and studied this during the top-down deployment of spatial attention in both perception and in working memory.

We considered four ways in which spatial biases in microsaccade directions and the N2pc might be related. First, microsaccades and the N2pc may be unrelated. Second, because microsaccades cause small shifts in retinal input that drive visual cortical activity [35,36], spatial biases in microsaccades may impose spatial modulations of the EEG. Third, the engagement of eye-muscles or corneo-retinal dipole movement during microsaccades may directly introduce artifacts [37,38] that may also lateralize according to microsaccade direction. Finally, microsaccade direction and the N2pc may both be modulated by a common underlying factor, without microsaccades *directly* causing the N2pc (as recently postulated for neural signatures relevant to attention [22,33]). Here, we provide evidence for a profound modulation of the N2pc by spatially biased microsaccades that is most consistent with the fourth scenario.

## Results

Healthy human volunteers participated in two complementary attention tasks that either required covert selection of one of two visual objects displayed to the left or right on the screen (Fig 1a), or internal selection of either of these visual objects from working memory (Fig 1h). In both cases, attentional selection was prompted via a central, nonspatial, color cue. Participants were required to compare the cued

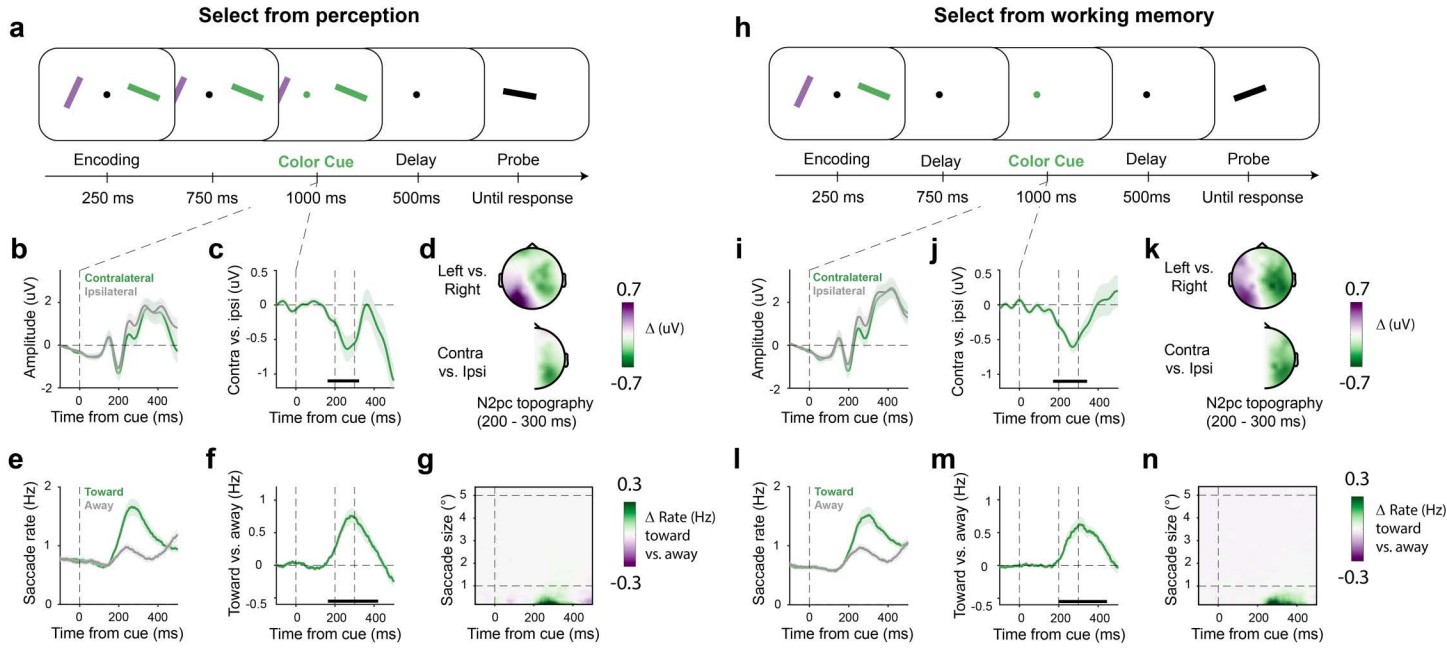

**Fig 1. Attentional selection evokes an N2pc and biases microsaccade directions at overlapping time windows in both perception and working memory.** Participants performed two tasks requiring covert selection of visual objects from perception (**a–g**) or working memory (**h, i**) following a central color cue. (**a, h**) Task schematics. In both versions of the task, an imperative central color cue indicated which visual objects had to be selected to compare its orientation to the upcoming probe stimulus. (**b, j**) Grand-averaged event-related potential (ERP) waveforms in posterior electrodes PO7/8 contralateral and ipsilateral to the side of the cued object. (**c, j**) The N2pc waveform reflecting the difference between contralateral and ipsilateral ERPs. (**d, k**) The topographies in the top show the voltage difference between trials where the left object was attended vs. trials where the right object was attended, across all electrodes, in the N2pc time window. The bottom topographies show the difference between contralateral and ipsilateral activity relative to the attended side, collapsed into the right electrode of all symmetric electrode pairs. Green shows stronger contralateral negativity signaling the N2pc. (**e, l**) Time courses of saccade rates (number of saccades per second) for saccades toward and away from the side associated with the cued object. (**f, m**) The spatial saccade bias, plotted as the difference between the rate of toward and away saccades. (**g, n**) Difference in saccade rates (toward minus away) as a function of saccade size. For reference, dashed horizontal lines indicate a common threshold for microsaccades (1° visual angle), as well as the center location of the selected object at 5° visual angle. All time courses show mean values, with shading indicating ±1 SEM calculated across participants (*n* = 23). Black horizontal lines in the time course plots indicate the significant temporal cluster (cluster-based permutation [39]).

(color-matching) visual object to an ensuing centrally presented black probe stimulus, judging whether it was rotated clockwise or counterclockwise. Note how in this set-up the use of the cue was imperative: without using the cue participants would not know to which visual object the probe should be compared. Participants were well able to perform both the perceptual (reaction time: 908 ± 60 ms (M ± SE); accuracy: 90 ± 1%) and working-memory (reaction time: 922 ± 61 ms; accuracy: 84 ± 1%) versions of this selective attention task.

## Attentional selection evokes an N2pc and a spatial microsaccade bias at overlapping time windows in both perception and working memory

Following the cue to covertly select the cued visual object from perception, we were able to establish the classic N2pc marker of spatial attention shifts: ERPs were more negative contra-compared to ipsilateral to the side of the cued object (Fig 1b), as quantified in predefined N2pc electrodes (PO7/PO8). Fig 1c zooms in on the spatial N2pc, displayed as a difference wave. Cluster-based permutation analyses confirmed a clear spatial modulation (cluster *P* < 0.001), with a cluster ranging from 162 to 323 ms after cue onset. Consistent with a modulation of visual-spatial attention, this difference wave had a predominantly posterior topography (Fig 1d).

In a similar time window where we found N2pc, we also observed a robust bias in the direction of eye movements according to the side of the to-be-attended stimulus (Fig 1e–1g). Following the cue, we found a higher rate of saccades toward versus away from the cued object (Fig 1e). The spatial saccade bias, again quantified as a difference wave (Fig 1f), was also highly significant (cluster $P < 0.001$) with a cluster ranging from 164 to 421 ms after cue onset. By visualizing this spatial bias as a function of saccade size (Fig 1g), we confirmed that this bias was driven almost exclusively by small saccades (<1°) in the classical microsaccade range (objects in the display were centered at 5°), consistent with prior studies during covert and internal shifts of attention [19–22,26]. This minute nature of the saccades driving this bias is one reason why such biases are easily overlooked in studies on the N2pc.

Highly similar results were found in the complementary task where the cue came *after* the visual objects were no longer visible—therefore prompting attentional selection within the spatial layout of working memory (Fig 1h–1n). Following the retro-cue, we again found a clear N2pc (cluster $P < 0.001$) (consistent with [6,11]) whose significant cluster ranged from 170 to 343 ms (Fig 1j), and that again had a posterior topography (Fig 1k). We also again found a clear spatial saccade bias (cluster $P < 0.001$) whose significant cluster ranged from 198 to 445 ms (Fig 1m), and that was again driven by eye movements in the microsaccade range (Fig 1n).

In both tasks, we thus observed not only clear N2pc EEG components associated with top-down shifts of spatial attention, but also robust spatial biases in microsaccades that co-occur in time. Having established this vital starting point, we now turn to our core question: how are these two markers of spatial attention related?

## Microsaccades strongly modulate but are not a prerequisite for the N2pc EEG marker of top-down covert and internal shifts of spatial attention

To investigate the dependence of the N2pc on the co-occurring spatial biasing of microsaccades, we adopted an approach also used and outlined in [22]. We leveraged the fact that, unlike continuous EEG signals, microsaccades are discrete events that can be classified at the single-trial level. To sort trials into relevant microsaccade classes, we defined the relevant attention window as the time window from 150 to 400 ms after cue onset (we chose this window to encompass microsaccades that slightly preceded or followed the N2pc). Having set this window, we defined three classes of trials: trials with (1) a microsaccade in the direction of the cued visual object ("toward-microsaccade trial"), (2) a microsaccade in the opposite direction ("away-microsaccade trial"), or (3) no discernible microsaccade anywhere in this key time window of interest ("no-microsaccade trial").

This analysis confirmed a dominance of toward versus away microsaccades (consistent with the spatial bias reported and quantified above), but also revealed a substantial proportion of trials in which no discernable microsaccade was detected in the window of interest (Fig 2a and Fig 2e). During perceptual selection, 37.5±2% (M±SE) of trials were classified as toward-microsaccade trials, 23.9±3% as away-microsaccade trials, and 38.6±3% as no-microsaccade trials. Similarly, during working-memory selection, 34.9±2% of trials were classified as toward-microsaccade trials, 24.5±1% as away-microsaccade trials, and 40.7±3% as no-microsaccade trials. This enabled us to separately examine the N2pc as a function of microsaccade class, with ample trials in each class.

When selecting from perception, trials with a toward microsaccade showed by far the clearest N2pc (Fig 2b green line; cluster $P < 0.001$), with a posterior topography (Fig 2d). In contrast, no N2pc cluster was found in neither the away-microsaccade nor in the no-microsaccade trial classes (Fig 2b, purple and gray lines).

To increase sensitivity, we also zoomed in the a-priori defined N2pc time window from 200 to 300 ms after cue onset (Fig 2c)—a window consistent with ample prior studies reporting an N2pc in this time range [5,7–10,12–15,40] as well as with the data from our tasks (see Fig 1c and 1j). A one-way ANOVA revealed a strong effect of microsaccade type ($F(2, 44) = 6.92$, $P = 0.002$, partial $\eta^2 = 0.239$). When considering individual conditions, we confirmed a highly robust N2pc in toward-microsaccade trials ($t(22) = -4.82$, $P < 0.001$, $d = -1.01$) and now also recovered a weak but significant N2pc in the no-microsaccade trials ($t(22) = -2.12$, $P = 0.046$, $d = -0.44$). However, even after averaging across the relevant N2pc time

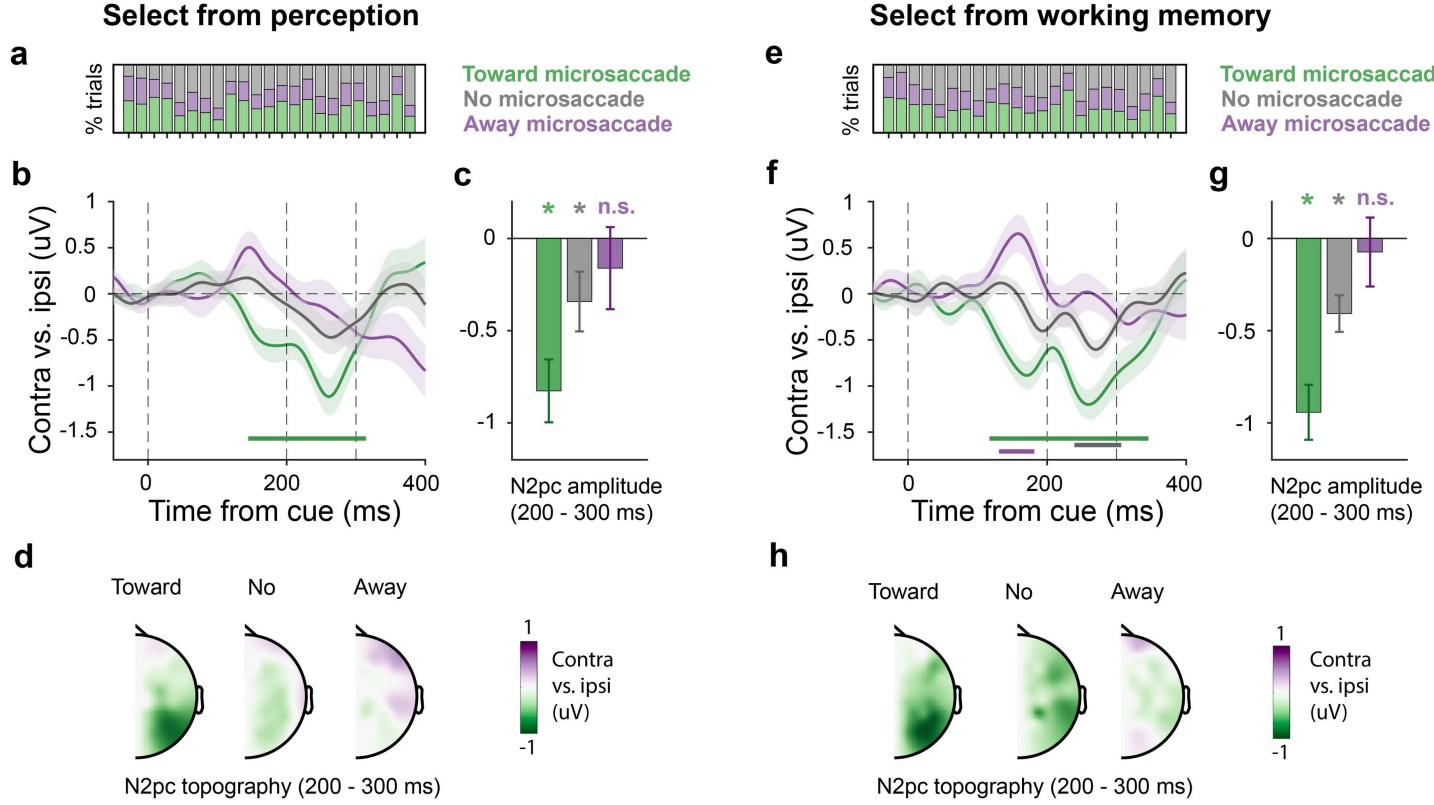

**Fig 2. Microsaccades strongly modulate but do not cause the N2pc EEG marker of top-down covert and internal shifts of spatial attention.**
Trials were classified into "toward-microsaccade," "away-microsaccade," and "no microsaccade" classes based on the presence/absence and direction of microsaccades in the relevant predefined 150–400 ms time window. (**a, e**) Participant-specific proportions of trial classes during selection from perception (panel **a**) and working memory (panel **e**). (**b, f**) N2pc waveforms (contra vs. ipsilateral ERPs in PO7/8) for the three microsaccade-based trial classes. Shading indicates ±1 SEM calculated across participants (n=23). Colored horizontal lines in the time course plots indicate significant temporal clusters (cluster-based permutation). (**c, g**) N2pc amplitudes in the three microsaccade trial classes extracted over the predefined 200–300 ms N2pc time window. The bar graphs show mean values, with error bars indicating ±1 SEM calculated across participants (n=23). The stars above the bars indicate that N2pc amplitude for that specific trial class is significantly different from zero (P<0.05). (**d, h**) N2pc topographies across trial classes plotted as contralateral vs. ipsilateral activity.

window, we still could not establish a significant N2pc in away-microsaccade trials (t(22) = −0.724, P=0.447, d=−0.15). Direct comparisons further confirmed that the N2pc amplitude in toward-microsaccade trials was significantly larger than in both no-microsaccade (t(22) = −3.12, $P_{Bonferroni}$=0.015, d=−0.65) and away-microsaccade trials (t(22) = −3.07, $P_{Bonferroni}$=0.017, d=−0.64). No significant difference in N2pc amplitude was observed between no-microsaccade and away-microsaccade trials (t(22) = −1.02, $P_{Bonferroni}$=0.96, d=−0.21).

Note how these differences in N2pc across the three microsaccade classes are unlikely accounted for by mere differences in the number of available trials. For example, while the N2pc more than halved in size when moving from toward- to no-microsaccade trials, we had in fact more (not less) trials in the latter condition.

A strikingly similar pattern of results was found when the cue prompted selection from working memory (Fig 2e–2h). We again found the clearest N2pc in toward-microsaccade trials (cluster P<0.001). This time, we also found a significant N2pc-cluster in the no-microsaccade trials (cluster P=0.005). However, in the away-microsaccade trials, we only observed an earlier cluster (cluster P=0.03) associated with a reversed ERP lateralization. When again zooming in on the N2pc time window—our key focus here—(Fig 2g), we again found a strong main effect of microsaccade trial-class

($F(2, 44) = 7.86$, $P = 0.001$, partial $\eta^2 = 0.263$). When considering individual conditions, we again observed a robust N2pc in toward-microsaccade trials ($t(22) = -6.31$, $P < 0.001$, $d = -1.32$) and a smaller but still robust N2pc in the no-microsaccade trials ($t(22) = -4.11$, $P < 0.001$, $d = -0.86$). However, just like we reported during perceptual selection, also during mnemonic selection, we could not establish a significant N2pc in away-microsaccade trials ($t(22) = -0.39$, $P = 0.7$, $d = -0.08$). Direct comparisons again showed that N2pc amplitude in toward-microsaccade trials was significantly larger than in both no-microsaccade ($t(22) = -2.92$, $P_{Bonferroni} = 0.024$, $d = -0.61$) and away-microsaccade trials ($t(22) = -3.22$, $P_{Bonferroni} = 0.012$, $d = -0.67$). Again, no significant difference was observed between no-microsaccade and away-microsaccade trials ($t(22) = -1.66$, $P_{Bonferroni} = 0.33$, $d = -0.35$).

For completeness, we also directly compared the microsaccade-related N2pc modulations between the perceptual and working-memory tasks (S1 Fig). We observed highly comparable N2pc amplitudes in both tasks for each microsaccade class (all $Ps > 0.5$), suggesting a similar modulation in both tasks. Thus, the observed N2pc modulations by microsaccades hold regardless of whether microsaccades bring attended visual targets physically closer to the fovea (as in perception) or not (as in working memory).

## The lack of an N2pc in the away-microsaccade trials is not due to a lack of attentional deployment

To ensure that the absent N2pc in the away-microsaccade trials was not due to a general failure to shift attention in these trials, we also examined two additional aspects of our data: behavioral performance and the lateralization of 8–12 Hz alpha-band activity which is another canonical neural marker of spatial attention shifts [22,41–43]. First, participants responded accurately (all conditions >80% accuracy) and quickly (all conditions <1,000 ms) across toward, away, and no-microsaccade trials in both the perceptual-selection and working-memory selection tasks (S2 Fig). If participants simply did not shift attention in the away-microsaccade trials, we should have seen strongly impoverished performance (the imperative cueing procedure we used necessitated using the cue to perform the task). Second, we observed preserved alpha-band lateralization reflecting spatial attention shifts in all trial classes (Fig 3; even if alpha-activity too was modulation by microsaccade presence and direction as in [22], see S3 Fig). This provides direct evidence that spatial attention

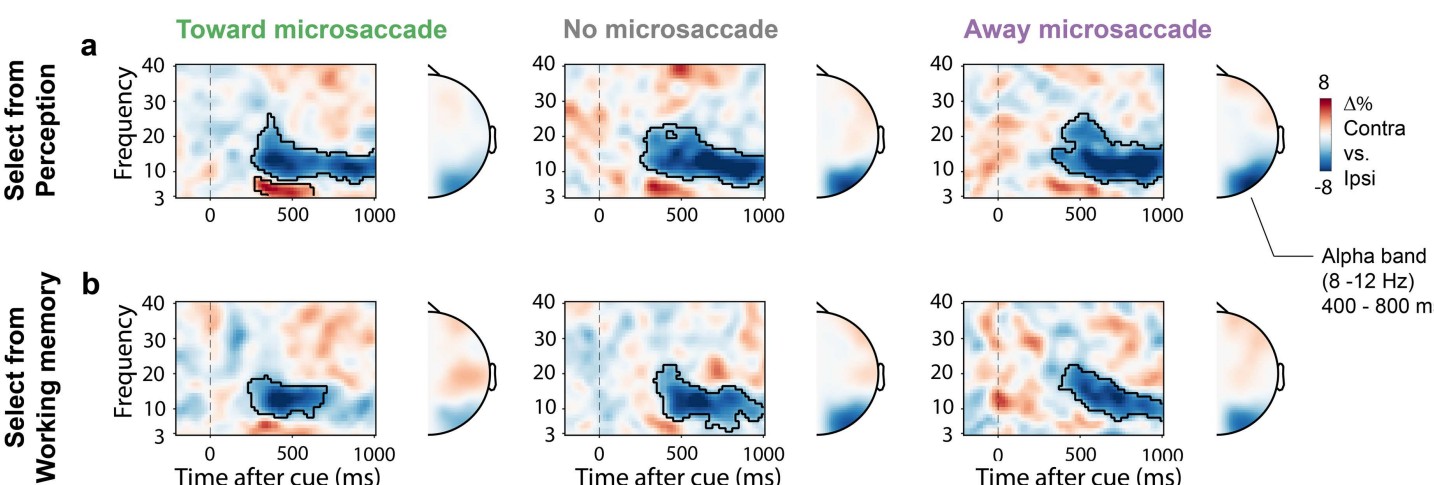

**Fig 3. The spatial lateralization of alpha activity is clear even in the away-saccade trials despite the lack of an observable N2pc in these trials.** (**a, b**) Time-frequency maps showing lateralization of EEG activity according to the side associated with the cued visual objects when selecting from perception (panel **a**) or working memory (panel **b**). We used the same electrodes as the ones used to quantify the N2pc (PO7/8). Topographies show the lateralization of 8–12 Hz alpha-band activity in the window from 400 to 800 ms after cue onset (as in [22]).

was also deployed in away microsaccade trials, even if we could not establish an N2pc in these trials. These data further show how the lack of an N2pc in these trials is not simply owing to a lack of sensitivity or poor EEG signal in these trials.

## Microsaccades do not directly cause the N2pc

In a final analysis, we investigated the temporal correspondence of microsaccades and the N2pc. We reasoned that if the observed N2pc modulation is driven by a direct consequence of microsaccades—due to changes in retinal input, eye-muscle activity, or corneo-retinal dipole movement—then the N2pc should precisely align with microsaccade onset across trials. Indeed, though the N2pc is typically considered as an attentional ERP component that is aligned in time to the onset of attentional cues and/or target stimuli, it remains possible that the N2pc is more aligned to the precise timing of the corresponding attentional modulation in microsaccades. To investigate this, we focused on trials that contained toward microsaccades and sorted them by microsaccade onset latency (using the logic of the ERP-image [44]). To increase the sensitivity and robustness of the analysis, we pooled the data from the perception and working-memory tasks (though the results are consistent when analyzed separately for both tasks; S4 Fig).

We first confirmed the validity of our microsaccade sorting. As shown in Fig 4a, the EOG showed a clear modulation that was precisely aligned to microsaccade onset. This shows that our sorting worked and that our eye-tracking measurements (from which we extracted microsaccades) and EEG measurements (that contained the EOG channel) were properly synchronized. Similarly, when looking at the non-lateralized ERP in electrode Oz (Fig 4b), we found a clear Lamda response (as in [35]) approximately 100–150 ms after microsaccades that also scales with microsaccade timing.

Critically, however, no such precise temporal alignment was observed for the N2pc—the blue negativity in Fig 4c that remains consistently aligned to the moment of the cue, emerging around 200–300 ms after cue onset. Indeed, for trials with late toward microsaccades, the N2pc often even preceded microsaccade onset. Thus, even though microsaccade presence and direction matter a lot for the N2pc (Fig 2), the N2pc is not directly locked in time to microsaccade onset. This rules out that the observed link between microsaccades and the N2pc is due to retinal displacement, eye-muscle artifacts, or corneo-retinal dipole movement, as such variables would have been expected to yield a precise locking in time (just like we had seen for the EOG and Lambda activity; Fig 4a and 4b). For a comprehensive overview across toward-, away-, and no-microsaccade conditions, see also S5 Fig.

## Discussion

As our data make clear, voluntary shifts of spatial attention in perception and working memory are not only associated with the N2pc (as also in [5–14]) but also with spatial biases in microsaccades (as also in [19–22,26]) that co-occur in time. We

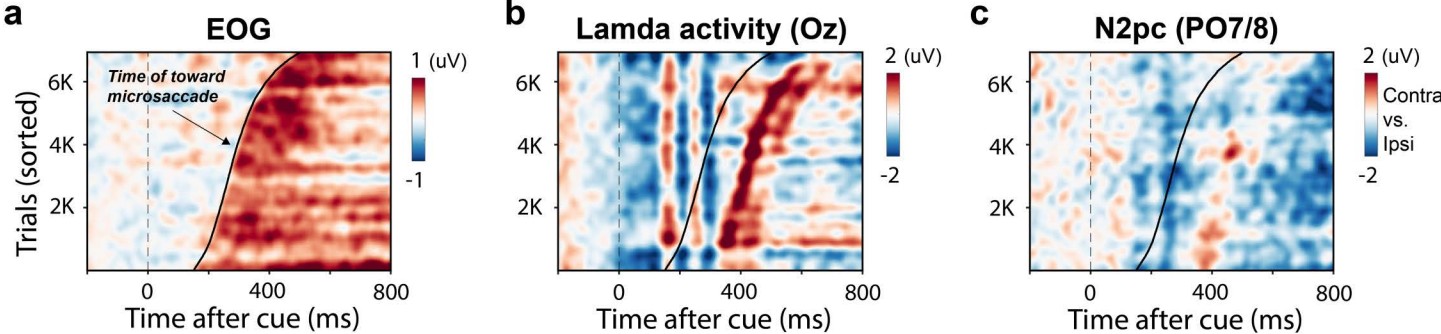

**Fig 4. EOG, Lambda, and N2pc activity in toward-microsaccade trials, sorted by the time of microsaccade onset.** Trials containing toward microsaccades were sorted by the latency of microsaccade onset (black line) and plotted as event-related potential (ERP)-images for horizontal EOG (**a**), Oz (**b**), and the lateralized N2pc in PO7/8 (**c**). Color indicates voltage (μV). Data were aggregated across all participants and both tasks.

now further show how co-occurring microsaccades profoundly modulate N2pc amplitude during the top-down deployment of spatial attention (e.g., the N2pc fully disappeared in trials in which microsaccades went in the direction away from the attended object). At the same time, we reveal how a severely weakened N2pc can still be observed in the absence of co-occurring microsaccades (showing microsaccades are not a *prerequisite* for the N2pc), and how the N2pc does not lock in time to the exact moment of microsaccades (showing microsaccades do not *directly cause* the N2pc).

We thus show that microsaccades do not *cause* the N2pc: the N2pc can still be observed in the absence of co-occurring microsaccades, albeit weak. This resonates with recent studies on complementary neural markers of attention that also reported that microsaccades are not a necessary prerequisite for neural modulations by attention to occur (cf. [22,33,34,36,45]; see also [32]). At the same time, we unveil that co-occurring microsaccades matter a lot for the N2pc: the N2pc more than doubles in size when microsaccades co-occur in the attended direction, while the N2pc vanishes when microsaccades co-occur in the opposite direction. This makes clear how microsaccades cannot be considered irrelevant when studying this popular human-neuroscience marker of attention. Admittedly, we studied the N2pc and microsaccades in relatively simple tasks that (1) enabled us to match the perceptual and working-memory selection tasks and (2) we knew would yield both markers. Prompted by our findings, it will now be vital for the field to adopt similar analyses in complementary tasks in which the N2pc is frequently reported.

One interpretation for the reported reduction and even lack of the N2pc in trials with no or away microsaccades is that attention may simply not have been deployed to the same extent in these trials. Additional features of our dataset counter this interpretation. First, because of the way we designed the task, if participants did not use the cue, they would have had no way of knowing which object was tested. This would have resulted in a large performance drop in these trials. We found no evidence for this. Second, a complementary marker of spatial attention—the lateralization of posterior alpha-band activity [22,41–43]—was observed in all three microsaccade trial classes. Critically, this marker was robust even in the away-microsaccade trials, where we no longer found an N2pc. These data suggest that the lack of an N2pc in the away-microsaccade condition is unlikely due to a lack of attentional deployment per se. Further note how the microsaccade-class-specific differences in N2pc amplitude are also unlikely accounted for by differences in trial numbers. For example, while the N2pc more than halved in size when moving from toward- to no-microsaccade trials, we had in fact more (not less) trials in the latter condition.

Despite the strong effect of microsaccade presence and direction on the N2pc, our ERP-image analysis (where we sorted the N2pc according to the timing of toward-microsaccades), showed that the N2pc did not align to the precise timing of microsaccade onset. This temporal dissociation suggests that microsaccades do not *directly* cause and/or contaminate the N2pc through low-level mechanisms like retinal displacement, eye-muscle activity, or corneo-retinal dipole movement, which would both be expected to be precisely locking in time to microsaccade onset. Instead, our findings support an interpretation whereby both the N2pc and microsaccades are driven by a common underlying process (e.g., a component of attention). In this interpretation, this common underlying process yields both the N2pc (whose timing aligns to the cue), and biases the direction of ongoing microsaccades (whose precise timings remain subject to additional factors like buildup of activity in the oculomotor system and refractory periods between eye movements [46,47]). This fits with the observation that attentional processes do not directly trigger new microsaccades, but only bias the direction of microsaccades that happen to be made around the time of the attentional shift [48]. In contrast, these same attentional processes may still directly trigger the N2pc. Such a scenario, though speculative, would yield a correlation between microsaccade directions (in the overall N2pc time window) and the N2pc, without a strict temporal correspondence at the level of single trials, as we observed here.

Our work complements prior studies linking microsaccades to neural activity [38,49–53] and to neural modulations by spatial attention [22,27–30,32–34,54]. We advance the latter literature by having uniquely targeted the N2pc, a canonical ERP marker of voluntary spatial attention from the human-neuroscience literature. While at least one prior study [34] also considered spatial ERP modulations to attention cues, ERP lateralization and microsaccade biases in that study reflected

PLOS Biology

bottom-up processing of the spatial cue, not top-down deployment of attention. Because we used a central color cue, any spatial modulation in our task must reflect top-down shifts of attention—the type of attention classically studied with the N2pc.

We not only considered attention in the context of perception (as in most prior studies) but also working memory. This provided not only an internal replication, but also a theoretically meaningful contrast: in perception, microsaccades bring attended visual targets physically closer to the fovea, whereas in working memory, they do not. While this contrast does not eliminate all visual motion across the retina, since other visual elements such as the fixation marker and screen borders remain, the difference in retinal displacement between these conditions would still be expected to be graded. Despite this, we found near-perfect generalization of our main findings across tasks. This suggests that retinal displacement alone is unlikely to fully account for the observed N2pc modulations, as this would have predicted stronger effects in the perceptual task, where more visual elements are displaced during eye movements. Our dissociation of precise microsaccade and N2pc timing provides further evidence against the contribution of retinal displacement to our findings.

Our interpretation that co-occurring microsaccade biases are not necessary for the N2pc to be observed hinges on how trustworthy the "no-microsaccade" condition is. We acknowledge that detecting microsaccades with perfect sensitivity and specificity remains a methodological challenge. Nevertheless, we implemented two critical steps to support the robustness of our interpretation. First, to minimize the risk of missing subtle microsaccades, we deliberately applied a low detection threshold (3 times the median gaze velocity), which we identified as the lower bound for producing stable ERP estimates (see S6 Fig). This conservative approach increases our confidence that trials labeled as "no microsaccade" are indeed free of eye movements in the critical analysis window. Second, we assessed the precise alignment of the N2pc to the precise timing of toward microsaccades. This revealed that, although microsaccade direction (in the overall N2pc time window) robustly modulated the N2pc, the N2pc was not precisely time-locked to the timing of microsaccade onset in single trials. This temporal dissociation suggests a more complex relationship between microsaccades and the N2pc than would be expected if the N2pc were merely an oculomotor artifact, and thereby provides further support that the N2pc is not directly driven by microsaccades.

We thus observed an indirect link between the N2pc and the brain's oculomotor system at the level of the peripheral output of the oculomotor system, as observed in microsaccades. Our findings leave unaddressed the relation between the N2pc and neuronal activity upstream in the brain's oculomotor system. Relevant oculomotor activity may remain below threshold for microsaccade execution (e.g., activity associated with unfulfilled microsaccades or microsaccade plans) but nevertheless plays a critical role in driving the N2pc. Accordingly, in future work, it will be important to supplement our approach with direct recordings in the superior colliculus (cf. [33,50]) to also assess the link between the N2pc and neuronal activity in circuits that govern microsaccade generation [50].

By measuring EEG, our data only allow us to conclude about N2pc presence and N2pc magnitude from the view of extracranial neural measurements. The fact that we did not observe an N2pc in away microsaccade trials at the scalp level, does not necessarily imply that no N2pc occurred anywhere inside the brain, as the N2pc may have been "masked" by additional signals imposed by the microsaccade. This possibility of "signal mixing" is hard to address based on the current data and likely requires intracranial measurements [55] or MEG [56], as well as additional types of analyses. This was beyond the scope of our current study, which was to delineate how much microsaccades contribute to the N2pc—both in perception and working memory—when following procedures that are customary in N2pc research. Potentially related to this point, we also noted an earlier lateralized EEG signal (~150–200 ms after cue onset) that preceded the classic N2pc time window and that varied with the direction of microsaccades, independent of side of the cued object (Fig 2b and 2f). This may possibly be related to the N170 response (for relevant prior work relating microsaccades to the N170, see [57,58]), though further work is needed to unpack this possibility.

Finally, while microsaccades may be considered a nuisance variable in neuroscience studies where instructed gaze-fixation yields fixed retinal positions of external visual inputs, the spatial microsaccade bias is also an interesting signal that brings opportunities for studying attention and working memory [26,59,60]. Like the N2pc, microsaccade biases also enable tracking of spatial attention shifts; they too have high temporal resolution, and they too track spatial attention shifts

in both perception [19,20] and working memory [22–25]. In future studies, it will therefore be interesting not only to systematically compare the inter-relation of both markers, but also their respective usefulness for studying covert and internal shifts of attention across different experimental tasks and settings. Future studies could further extend these approaches to other neural markers of attentional selection, and to other microscale oculomotor behaviors, like ocular tremor and drift.

## Methods

### Ethics

Experimental procedures were reviewed and approved by the scientific and ethical review board of the faculty of behavior and movement sciences at Vrije University Amsterdam (ID: VCWE-2020-155). This study was conducted according to the principles expressed in the Declaration of Helsinki. Participants provided written informed consent prior to the experiment and were compensated €10 per hour (or the equivalent in credits) for their time.

### Participants

Twenty-three healthy human volunteers participated in the study (age range: 19–32 years; 2 males and 21 females; 22 right-handed; 15 corrected-to-normal vision). The sample size of 23 was set a priori based on a prior study from the lab that addressed a similar research question [22] for a complementary neural signature. To reach the desired sample size, the first participant had to be replaced due to an error in the experiment code.

### Task design and procedure

Participants performed two complementary spatial-attention tasks that either required covert selection of one of two visual objects displayed to the left or right on the screen (select from perception) or internally select either of these visual objects that were no longer visible at the time of the attention cue (select from working memory; Fig 1h). We designed our tasks in such a way that they were as similar as possible, and only differed whether task performance relied on visual object selection from perception or from working memory.

Trials started with the presentation of two visual objects (bars with distinct colors and randomly drawn orientations) centered at 5° to the left and right of a central fixation dot. Bars were 2 by 0.4° visual angle, and the fixation dot had a radius of 0.07° visual angle. In the perceptual-selection task (Fig 1a), the stimuli stayed on the screen for 2000 ms. While the stimuli were still visible (1,000 ms after stimuli onset), the central fixation dot changed color for 1,000 ms, acting as a 100% valid attention cue. This cue instructed participants to select the color-matching target object for the ensuing test. After the offset of the stimuli, a blank display was presented for 500 ms, before the onset of the test display. This ensured that participants needed to select and remember the cued bar ahead of time. In the final test display, a black bar appeared at the center of the screen, rotated between 10 and 20° clockwise or counterclockwise relative to the orientation of the cued bar. Participants were required to indicate whether the cued object (meanwhile in memory) should be rotated clockwise or counterclockwise to match the black bar in the test display. Note how cues were imperative in this task, as the test display itself did not contain any information as to which bar was being tested. After responding, participants received immediate feedback indicated by a number ("0" for incorrect and "1" for correct) displayed for 250 ms just above the fixation point. Inter-trial intervals were randomly drawn from 500 to 1,000 ms.

In the working-memory-selection version of this attention task (Fig 1h), the procedure was identical except that the stimuli only briefly appeared at the start of each trial for 250 ms and thus were no longer visible when the cue was presented. Instead, the cue now instructed which colored object to select from working memory. The onset timings of the stimuli, the cue, and the test display were identical between the two tasks. Note also how both tasks eventually relied on a memory-guided report. The only and key difference was whether the cue prompted selection of a visual object that was currently visible (select from perception) or not (select from working memory).

In each trial, the bars were randomly assigned two distinct colors selected from a set of four: blue (RGB: 21, 165, 234), orange (RGB: 234, 74, 21), green (RGB: 133, 194, 18), and purple (RGB: 197, 21, 234). The orientations of the bars were randomly drawn from 0° to 180° with a minimum difference of 20° between them. Each bar was equally likely to be cued for report. Across trials, we made sure to equally often cue the left or the right object. During the test display, the black bar (RGB: 64, 64, 64) was oriented either clockwise or counterclockwise from the cued target object, with a random change in orientation ranging from 10° to 20°.

The experiment consisted of 3 consecutive sessions, each containing 10 blocks of 40 trials. Within each session, 5 blocks contained the perceptual-selection version of our task, and 5 blocks the working-memory-selection version, in random order. The task version was explicitly stated prior to each block. Each participant completed a total of 1,200 trials, evenly divided between the perceptual (600 trials) and working-memory (600 trials) attention tasks. Prior to the formal testing, participants practiced for approximately 10 min to familiarize themselves with both task types.

The experiment was programmed in Python (version 3.6.13) with Psychopy (version 2021.2.2). During the experiment, participants sat in front of a monitor (with a 100-Hz refresh rate) at a viewing distance of ~70 cm with their head resting on a chin rest.

## Eye-tracking acquisition and pre-processing

We tracked the one eye (which was by default the right eye) for all participants with an EyeLink 1000 system (SR Research) at a sampling rate of 1,000 Hz. The eye-tracker camera was positioned approximately 5 cm in front of the monitor, 65 cm away from the participant. Gaze position was continuously monitored along both the horizontal and vertical axes. Prior to recording, we used the built-in calibration and validation protocols of the EyeLink software. After recording, the eye-tracking data were converted from the original.edf format to.asc format and then analyzed in MATLAB using the FieldTrip analysis toolbox with custom scripts. Before turning to saccade-detection, we identified blinks by detecting 0 clusters in the gaze data, and then we set all data from 100 ms before to 100 ms after the detected 0 clusters to Not-a-Number (NaN) to remove any residual blink artifacts. Finally, the data were epoched from −1,000 ms to +1,500 ms relative to the onset of the attention cue.

## Microsaccade detection

To identify microsaccades, we employed a velocity-based detection method, which we extensively validated in our previous article addressing a similar question [22]. Because the objects were always horizontally arranged (one left, one right), our analysis focused on the horizontal channel of the gaze data, which is also the same as in our previously validated approach. In addition, we confirmed that the results remain consistent when using a 2D velocity vector that combines both horizontal and vertical gaze components for microsaccade detection (see S7 Fig).

We first calculated the gaze velocity by taking the Euclidean distance between temporally successive gaze-position values in the horizontal axis. To increase the SNR, we smoothed velocity by applying a Gaussian-weighted moving average filter with a 7-ms sliding window using MATLAB's built-in "smoothdata" function. We identified saccades when the velocity exceeded a trial-based threshold of 3 times the median velocity, marking the first sample that crossed the threshold as the onset of the saccade. We note how the default setting in our custom saccade-detection function is a threshold of 5 times the median gaze velocity. Doing so, we increase confidence that our "no-microsaccade" condition really did not contain any microsaccade (at the expense of the occasional misclassification of noise as a "toward-microsaccade" or "away-microsaccade" microsaccade). We adopted the exact same logic and threshold in [22]. To further validate this choice, we conducted additional analyses using a range of thresholds, confirming that a threshold of 3 times the median gaze velocity was a lower bound for retaining sufficient trials to yield interpretable ERPs (see S6 Fig). To prevent counting the same saccade multiple times, we imposed a minimum delay of 100 ms between successive saccades. The magnitude and direction of each saccade were determined by comparing pre-saccade gaze positions (−50 to 0 ms before saccade onset)

with post-saccade gaze positions (50 to 100 ms after saccade onset). We only considered gaze shifts with an estimated magnitude of at least 0.05 visual degrees (3 arcmin), because it is hard to ascertain the direction of the smaller gaze shift. Trials in which we observed gaze shifts smaller than 0.05° in the analysis window of interest were removed from further analysis.

We calculated gaze shift rates (measured in Hz) using a sliding time window of 50 ms, advanced in steps of 1 ms. To quantify the spatial biasing of saccades, we classified saccades as "toward" or "away" from the cued visual object (or its memorized location) and compared the rates of toward and away saccades. In addition, to verify the microsaccadic nature of the reported spatial saccade biases, we chose not to set an arbitrary threshold on saccade sizes, but rather to visualize the spatial saccade bias as a function of saccade size (as in [22]). For this, we decomposed shift rates into a time-size representation (Fig 1g and 1n). For saccade-size sorting, we employed successive saccade-size bins of 0.2 visual degrees, with increments of 0.04 visual degrees.

### Microsaccade-based trial sorting

A key step in our analysis was to separate the trials based on the presence/absence and direction of microsaccades occurring in the "attention window of interest". We set this window to 150–400 ms after cue onset. This ensured to capture the N2pc time window (from 200 to 300 ms post cue onset), while also including potential saccades that preceded or followed in time. If no gaze shift was detected in this key window, the trial was labeled as a "no-microsaccade" trial. If a gaze shift *was* detected, we further categorized the trial by the direction of the first detected shift in this window, classifying the trial as either a "toward-microsaccade" or "away-microsaccade" trial, in reference to the left/right location of the cued visual object.

Before trial classification, we excluded any trials where the eye-tracking data contained NaN values (typically due to blinks or temporary loss of eye signal) anywhere within attention window of interest (150–400 ms after cue onset), or trials in which the detected gaze shifts were smaller than 0.05°.

### EEG acquisition and pre-processing

We used the BioSemi ActiveTwo System (biosemi.com) with a conventional 10–10 System 64-electrode setup. Two electrodes on the left and right mastoids were used for offline re-referencing of the data. For the purpose of data cleaning, we also measured EOG, with two electrodes placed horizontally near the left and right eyes and two electrodes above and below the left eye. EOG measurements did not serve to track eye movements (for this, we used the dedicated Eyelink eye-tracker), but exclusively served as useful signals to identify Independent Component Analysis (ICA) components for removal.

We used MATLAB (2022a) to analysis EEG with the FieldTrip toolbox [61] and custom codes. We first epoched the data from 1,000 ms before to 1,500 ms after cue onset, re-referenced the data to an average of the mastoid electrodes. After that, we conducted an ICA and correlated the ICA components with recorded EOG to identify components for rejection. Finally, we removed the trials with exceptionally high variance using the function ft_rejectvisual in Fieldtrip. All the data cleaning was performed without knowledge of the experimental conditions to which individual trials belonged. All analyses were performed on data at a 1,024 Hz sampling rate.

### N2pc analysis

The epoched EEG data were first baseline corrected by subtracting the average potential in the 250-ms window preceding cue onset, and then averaged across trials for each condition. To extract the N2pc, we focused on activity in predefined posterior visual electrodes PO7/PO8 (as also in [8,15,40,62]) and calculated the difference between contralateral and ipsilateral waveforms, relative to the location (or memorized location) of the cued object. In addition to visualizing and

evaluating the full-time courses of this ERP lateralization, we also extracted the average N2pc amplitude by averaging over the predefined window of 200–300 ms after cue onset, based on ample prior reports of the N2pc in a similar time range [5,7–10,12–15,40]. Trial-averaged ERPs were smoothed using a Gaussian kernel with a standard deviation of 15 samples (~15 ms at our sampling rate of 1,024 Hz).

## Time-frequency analysis

Though our focus was on the N2pc (an ERP component characterized in the time domain), for completeness, we also considered the lateralization of 8–12 Hz alpha-band activity. To this end, we first applied a time-frequency decomposition using a short-time Fourier transform with Hanning-tapered data. A 300-ms sliding time window was used to estimate spectral power to estimate spectral power within the 3–40 Hz range in the step of 1 Hz, progressing in steps of 20 ms. To quantify lateralization, we used the same predefined posterior visual electrodes (PO7/PO8) that we used to extract the N2pc. Lateralization was again calculated by comparing activity contralateral versus ipsilateral to the location of the cued visual object. For spectral power, this contrast was expressed as a normalized difference: [(contra − ipsi)/(contra + ipsi)] × 100.

## ERP-image analysis

To assess the precise temporal relationship between the N2pc and microsaccade onset, we performed an ERP-image analysis [44], focusing on trials containing toward microsaccades. Trials from all participants were pooled and sorted based on the latency of the toward microsaccade in each trial (relative to cue onset). For visualization, the sorted 2D trial matrix (trials × times) was smoothed using a gaussian kernel (using MATLAB's built-in "smoothdata" function), applied first across 10% of neighboring trials in sorted order (y-axis in the 2D matrix) and then across a 50-ms slide window in the time domain (x-axis in the 2D matrix).

## Analysis of behavioral performance

We analysed both task accuracy (percentage correct responses) and response times (counted from probe onset). Response times were cleaned using a two-step trimming procedure. First, trials with response times exceeding 3,000 ms were excluded. Second, data were further trimmed using a cutoff of 2.5 standard deviations from the participant's mean. All behavioral performance analyses were conducted on the trimmed dataset.

## Statistical analysis

For statistical evaluation of the time-series data, we used a cluster-based permutation approach [39], that evaluates the reliability of neural and gaze patterns across neighboring time points while controlling for (i.e., effectively bypassing) multiple comparisons. First, a permutation distribution of the largest clusters was created by randomly permuting the trial-averaged condition-specific data at the participant level. We then calculated the $p$-value for each observed cluster by using the proportion of permutations where the largest cluster exceeded the size of the observed cluster. We performed 10,000 permutations and identified clusters using Fieldtrip's default settings (grouping adjacent same-signed data points significant in a mass univariate $t$ test with a two-sided alpha level of 0.05, and defining cluster size as the sum of all $t$ values within the cluster).

To statistically confirm that N2pc amplitude was modulated by microsaccade trial class (i.e., toward, away, and no microsaccade trials), we additionally employed a one-way ANOVA on N2pc amplitude averaged across the predefined N2pc time window. We supplemented this with post-hoc condition comparisons, and also compared the N2pc within each trial-type to zero, to test for evidence on the absolute presence/absence of the N2pc. We performed all analyses separately for the two versions of our task in which the cue prompted selection either from perception or from working memory. A direct comparison between tasks was beyond the scope of our central aims. We included both tasks only to seek

generalization across these two complementary task settings that have each been shown to yield both N2pc and micro-saccade modulations [5,6,11,19,20,22,26].

## Supporting information

**S1 Fig. Comparison between the N2pc from perceptual and working memory tasks that are separately shown in** Fig 2**.**
(TIF)

**S2 Fig. Behavioral performance as a function of microsaccade trial classes in the task requiring selection from perception (a, c) or working memory (b, d).** The mean of reaction time (**a, b**) and the mean of accuracy (**c, d**). Bar graphs show mean values, with error bars indicating ±1 SEM calculated across participants ($n = 23$). For reaction time, a one-way ANOVA did not yield an effect of microsaccade trial-class in the perceptual-selection task ($F(2, 44) = 0.2$, $P = 0.82$, partial $\eta^2 = 0.009$), but did show a main effect in the working-memory selection task ($F(2, 44) = 6$, $P = 0.005$, partial $\eta^2 = 0.21$). Post-hoc $t$ test in the latter task showed how reaction times in toward-microsaccade trials were shorter than in both no-microsaccade ($t(22) = -3.89$, $P_{Bonferroni} = 0.002$, $d = -0.81$) and away-microsaccade trials ($t(22) = -2.82$, $P_{Bonferroni} = 0.03$, $d = -0.59$). No significant difference was found between no-microsaccade and away-microsaccade trials ($t(22) = 0.4$, $P_{Bonferroni} = 1$, $d = -0.08$). For accuracy, one-way ANOVAs did not yield a significant effect of microsaccade trial-class, neither in the perceptual-selection task ($F(2, 44) = 1.26$, $P = 0.29$, partial $\eta^2 = 0.05$) nor in the working-memory selection task ($F(2, 44) = 0.28$, $P = 0.76$, partial $\eta^2 = 0.01$).
(TIF)

**S3 Fig. Complementary to the time-frequency analysis in** Fig 3**, we overlaid and compared the averaged 8–12 Hz alpha lateralization.** Black horizontal lines indicate significant temporal clusters for the comparisons indicated on the right (two-sided cluster-based permutation test). Time courses show mean values, with shading indicating 95% SEM (calculated across 23 participants).
(TIF)

**S4 Fig. Complementary to ERP-image analysis in** Fig 4 **by showing the same analysis split by perception and working-memory conditions.**
(TIF)

**S5 Fig. Complementing** Fig 4 **by showing the same analysis also in trials with away microsaccades and without microsaccades in the window between 150 and 500 ms after cue onset.** Note how, in contrast to toward- and away-microsaccades trials, trials without microsaccades have no reference event to sort the trials by and are here only included for completeness.
(TIF)

**S6 Fig. N2pc waveforms in trials classified as "no-microsaccade" under varying velocity thresholds.** We compared ERP results in the "no-microsaccade" condition using different microsaccade detection thresholds (2, 3, 4, or 5 times the median gaze velocity). Waveforms reflect the contralateral-minus-ipsilateral difference at PO7/8. The top row shows results for the "select from perception" condition; the bottom row shows results for the "select from working memory" condition. A threshold of 2 (times the velocity) results in unstable and noisy ERPs. Thresholds of 3 (times the velocity) and above yield consistent and interpretable N2pc effects. Shading indicates ±1 SEM across participants ($n = 23$).
(TIF)

**S7 Fig. Consistent results when detecting microsaccades using 1D (horizontal) versus 2D (horizontal and vertical) gaze data.** We re-ran our microsaccade-detection approach using a 2D velocity vector, combining horizontal and

vertical gaze positions by calculating the 2D Euclidean distance between temporally successive samples. The remaining steps in the detection pipeline (including smoothing, thresholding, onset identification, direction classification, and magnitude filtering) were kept identical to the original 1D method. (**a**) Time course of spatial bias in microsaccade direction, detected using either 1D (solid line) or 2D (dashed line) velocity-based methods. (**b**) N2pc waveforms (contra versus ipsilateral ERPs in PO7/8) for trials categorized by microsaccade direction (Toward target, away from target, or no microsaccade), using both detection methods. Shading indicates ±1 SEM calculated across participants ($n = 23$). Colored lines correspond to microsaccade detection using 1D data; gray dashed lines represent detection using 2D data.
(TIF)

## Acknowledgments

We thank Sisi Wang and Anna van Harmelen for their valuable input on the article.

## Author contributions

**Conceptualization:** Baiwei Liu, Freek van Ede.

**Data curation:** Baiwei Liu, Siyang Kong.

**Formal analysis:** Baiwei Liu, Siyang Kong.

**Funding acquisition:** Freek van Ede.

**Investigation:** Baiwei Liu, Siyang Kong, Freek van Ede.

**Supervision:** Baiwei Liu, Freek van Ede.

**Visualization:** Baiwei Liu.

**Writing – original draft:** Baiwei Liu, Freek van Ede.

**Writing – review & editing:** Baiwei Liu, Freek van Ede.

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
