## [Editor Report · Decision Letter 0]

20 Jan 2025

Dear Dr Liu, 

Thank you for submitting your manuscript entitled "Microsaccades strongly modulate but do not necessarily cause the N2pc marker of spatial attention" for consideration as a Short Reports by PLOS Biology.

Your manuscript has now been evaluated by the PLOS Biology editorial staff as well as by an academic editor with relevant expertise and I am writing to let you know that we would like to send your submission out for external peer review.

Once your full submission is complete, your paper will undergo a series of checks in preparation for peer review. After your manuscript has passed the checks it will be sent out for review. To provide the metadata for your submission, please Login to Editorial Manager (https://www.editorialmanager.com/pbiology) within two working days, i.e. by Jan 22 2025 11:59PM.

Kind regards,

Christian

Christian Schnell, PhD

Senior Editor

PLOS Biology

cschnell@plos.org

---

## [Decision Letter · Decision Letter 1]

19 Mar 2025

Dear Dr Liu,

Thank you for your patience while your manuscript "Microsaccades strongly modulate but do not necessarily cause the N2pc marker of spatial attention" was peer-reviewed at PLOS Biology. It has now been evaluated by the PLOS Biology editors, an Academic Editor with relevant expertise, and by several independent reviewers. 

In light of the reviews, which you will find at the end of this email, we would like to invite you to revise the work to thoroughly address the reviewers' reports.

As you will see below, the reviewers agree that the study addresses an important topic. Most of the concerns they raise can likely be addressed through more careful discussion of the literature and additional analyses.

Given the extent of revision needed, we cannot make a decision about publication until we have seen the revised manuscript and your response to the reviewers' comments. Your revised manuscript is likely to be sent for further evaluation by all or a subset of the reviewers.

**IMPORTANT - SUBMITTING YOUR REVISION**

*Re-submission Checklist*

*Published Peer Review*

*PLOS Data Policy*

*Blot and Gel Data Policy*

Sincerely,

Christian

Christian Schnell, PhD

Senior Editor

PLOS Biology

cschnell@plos.org

REVIEWS:

Reviewer #1 (Leon Y. Deouell): The manuscript by Liu, Kong and van Ede examines whether the N2pc, a commonly used lateralized EEG measure of spatial attention, depends on the occurrence of microsaccades, small saccades occurring involuntarily while attempting fixation. The question is appropriate - several studies in the last decade or so have shown that while attempting to maintain fixations (i.e. when subjects are required to not move their eyes), microsaccades are nevertheless biased in the direction of spatial attention. Moreover, several previous studies have shown that EEG measures (ERPs and spectral components) can be affected by the occurrence of microsaccades. 

Here, Liu et al. presented subjects with cued spatial attention tasks, requiring the subjects to maintain fixation while awaiting a probe relating to either the left or right stimulus. A central cue indicated which side is to be probed. An interesting twist was that in one condition the left and right stimuli remained on the screen when the side to be attended was cued, while in another condition, the stimuli were gone by the time the cue appeared, requiring a shift of spatial attention over memoranda. As expected based on the above mentioned studies, microsaccades were strongly biased towards the cued side. The N2pc was robust when subjects made a saccade towards the attended side, and absent when they made a saccade in the other direction, suggesting a strong modulation. The critical point is what happens in trials in which no saccade was made (a bit over a 1/3 of the trials). Here, the authors argue that there is still a small but significant effect, which drives them to the main argument reflected in the manuscript title. 

Overall, this study presents an important cautionary note about the importance of careful examination of eye movements in EEG experiments, especially when lateralized stimuli are presented (but not only, see Yuval-Greenberg et al., 2008, 2009; Dimigen et al., 2009). The results are quite striking (but see questions about the interpretation below) and the inclusion of both the perceptual and the memory-based task is elegant.

Major comments:

1. The robust effect in this study is that absent a microsaccade in the direction of the attended side, the N2pc is greatly reduced, and thus the argument for the residual effect requires scrutiny. This is especially the case since in one of the conditions (the perceptual one), the N2pc is feeble, and is not significantly different from the away-microsaccade effect. This requires careful examination of the processing stages to determine if there were truly no microsaccades in the so-designated bin:

a. Was only the right eye measured? The Eyelink 1000 allows for a binocular measurement, and especially for microsaccades, the correlation between the eyes is relevant (See Kliegel's algorithm). If only the right eye was actually measured, was it the dominant eye for all subjects? 

b. What happens if both the horizontal and vertical signals are used when calculating velocity, instead of only the horizontal? Does it make a difference?

c. The permissive velocity threshold seems like a justified conservative decision. But how do the results change if you change the velocity threshold up or down (a multiverse result would be helpful to determine the effect of such decisions)?

2. I missed an attempt to explain how or why the microsaccades affect the N2pc measurement. In principle, this can happen in at least 3 ways - 1) the change in retinal image caused by the microsaccade can change visual cortical responses 2) the electric potential caused by the movement of the eye and the activity of ocular muscles could confound\distort the desired signal in several ways (see Keren et al., 2010), and 3) the more benign reason for a correlation between microsaccades occurrence and ERPs is that there is a common factor driving both, namely, attention. The authors argue based on the behavioral and alpha results that attention was similarly shifted in trials with no saccades, basically arguing against cause #3. The memory-based condition mitigates cause #2 (lines 262-264) although it should be noted that the cue remains on the screen and becomes lateralized when a microsaccade occurs. This leaves us with option #2. So, is it a matter of signal mixture that makes the attentional N2pc seem larger when microsaccades co-occur? Perhaps a trial-by-trial analysis like the one in Figure 4 of Dimigen et al., 2009 (J Neurosci 29, 12321-12331) or Figure 4 of Yuval-Greenberg et al., 2008 could help?

3. Figure 2 reveals an early effect (~150-200ms) where "Toward" and "Away" saccades create opposite asymmetries - that is, stronger negativity contralateral to the direction of saccades, rather than the direction of cued attention. There is a trend in the perception condition and significant effects in the memory condition (in the memory condition this trend is also seen in the 200-300 window by the pinkish blob in the occipital half-scalp in the Away condition). This effect may also be clarified by the trial by trial analysis suggested above.

Minor comments

4. A discussion of the difference between the perceptual and memory conditions is warranted. What could be the reason for the finding of N2pc with no microsaccades in the memory condition and not so much in the perceptual condition (in the same subjects)?

5. It is up to the editors to decide, but I would prefer to have the supplementary figures in the main text. They are important enough in my opinion

Reviewer #2: This is a nice and clearly written paper with nice graphics. It also touches on an important topic.

I have some comments below, primarily related to placement in the proper context of the literature, and also with suggestions to avoid getting bogged down by some unnecessary debates:

- Lines 26, 27: what if there are unrealized microsaccade programs? i.e. motor preparation (this is elaborated on a bit in a later comment)

- Line 43: Microsaccades are not involuntary! See Willeke et al., Nat. Comm., 2019, which also explains the long history of this misconception

- Lines 48, 49: It should be added that this is a relevant point and not just theoretical, especially given work like: Hafed, Neuron, 2013; Hafed et al., Front. Sys. Neurosci, 2015; Tian et al., Front. Sys. Neurosci., 2016; Chen et al., Curr Biol, 2015; Zhang et al., BioRxiv, 2024. I.e. this is something that already has some support, and therefore warrants further investigation in this study and future ones

- Line 53, for the references 22, 27, 28, there's also Chen et al., Curr Biol 2015

- Lines 56-58: is this really the case? If the memory representation is embedded in visual maps, then a retinal refresh could still differentially modulated an embedded memory representation. Indeed, eye movements strongly modulate sensory maps even in the absence of sensory drives. So, if there's something else embedded in these maps, then they can be modulated

- The legend of Fig. 1d (and the main text) are too terse and sparse about what is being shown. Since this is a multi-disciplinary journal, what is written and shown in Fig. 1d is currently simply incomprehensible for people outside the field. For example, what is the purple and what is the green? And how is this indication of "posterior" specificity? And what is the difference between left vs. right and contra vs. ipsi? And, why is left so purple? And what is the color bar in the first place? Too many details are completely missing. Panels b and c are comprehensible, but d is just too abstract. Similarly for k

- Line 200: I'm a bit confused here. You're using a brain signal as the "objective" measure ("marker") that attention was deployed. But, the current paper is already showing clearly that this is dangerous, since the N2PC component is clearly microsaccade-related and practically non-existent with attention shifts. Thus, the N2PC "marker" was already proven to not be a marker, and this could also be the case for other markers. So, isn't this circular logic????

- For me, Fig. 2 is convincing that it's essentially all explained by the towards microsaccades. The small cluster that emerges in the memory condition is qualitatively drastically very different from the classic N2PC, e.g. later, and oscillatory, etc etc etc. I understand that the authors are treading a bit carefully, but I see from this paper that basically all of the N2PC effect is explained by microsaccades, and not just "most". The stuff happening later with very different dynamics (e.g. oscillatory) etc could be explained by other things completely, including (but not exclusively) unrealized motor plans

- Line 226: really? See the above comment. Also, there's always the idea that there was a microsaccade plan that was still not realized. I think that this is a distinct possibility. I know that this sounds like I'm somehow rooting for microsaccades. However, this is exactly the kind of argumentation that people rooting for attention use in order to discount an important role for microsaccades in attention. So, if they can do it there, then it can also be done here. Of course, I don't mean to just do it for the sake of doing it (see later comment about "threat"). What I mean is to consider all possibilities with an open mind

- Line 228: this is not the complete story with these citations. For example, see the Hafed lab results mentioned above. Also see the latest Hafed lab work on this topic: Zhang et al., BioRxiv, 2024. In that work, they seem to try to make sense of all of this fuss in their discussion.

- Line 243: again, this "marker" is problematic. What if someone later finds that it's also "severely weakened" by oculomotor activity? I would much rather see mechanistic explanations of how things happen in a brain as opposed to sticking to labels like attention, but that's just me. I mean, no one would ever debate that eye movements strongly modulate brain activity. So, how come we suddenly have to shy away from this when the eye movements are small???? The authors might have an opportunity to make this point clearer here in this manuscript

- Line 276: if someone views a brain phenomenon as a "threat", then they simply should not be brain scientists at all. Why diminish the good work in this paper by getting into such pettiness? I would much rather make this text as a marveling on the revelations that are allowed by the experiments, and not getting too apologetic because some people feel "threatened". Really, I mean, is science about just protecting one's own work (and job) or about learning something new?

- One small minor thing that can be mentioned: there is now increasing evidence that ocular position drifts can also reflect stimulus locations and properties. So, it might be nice to add a sentence or two somewhere indicating that a future direction would be to consider the even smaller ocular position drifts. Who knows, maybe they will also account for the remaining weak n2pc in the absence of microsaccades.

Reviewer #3 (Shlomit Yuval-Greenberg): This is an valuable study examining the relationship between microsaccades and the N2pc marker of spatial attention. The findings highlight the temporal co-occurrence of microsaccades and N2pc and demonstrate that microsaccades significantly modulate the EEG response. Crucially, the study also claims that microsacacdes are not a prerequiste for the N2pc, suggesting that this component is not an artifact of eye movements.

I find this topic exceptionally relevant and timely. EEG is becoming increasingly popular as a first-choice brain imaging method due to its accessibility and relatively low cost. However, EEG is also highly susceptible to artifacts, including those from eye movements. Demonstrating both the limitations and advantages of the method is a valuable contribution of this study. Therefore, I support its publication in PLOS Biology, but I have a few concerns and believe the paper could be improved.

Major Concerns:

1. The paper could provide a more in-depth discussion of the possible mechanisms linking microsaccades and N2pc, supported by additional analyses. Several potential explanations exist, including: Corneo-retinal dipole artifacts, the spike potential artifact (see Yuval-Greenberg et al., 2009), visual cortical activity (Lambda wave) (see Olaf Dimigen's studies), non-visual activity (such as attentional shifts, oculomotor-related processes, and decision-making processes).

A more thorough review of these possibilities in the introduction, along with a discussion of how the current findings fit within this framework, would strengthen the paper. Additionally, ruling out some of these explanations could be supported by further analyses, particularly examining the topographical distribution of the effect. It would be valuable to see whether the ERP topography differs in the presence vs. absence of microsaccades and whether these differences align with known microsaccade-related artifacts.

2. The authors claim that the N2pc observed in the working memory condition cannot be attributed to a visual response (Lambda wave) because there is no retinal displacement of the attended object. However, this assumption is problematic. Even without a stimulus on the screen, microsaccades still cause peripheral retinal slips. Given that the experiment was not conducted in total darkness, light from the monitor, its frame, and other elements in the experimental room could produce substantial retinal shifts, leading to visual responses.

This possibility could be ruled out by conducting topographical analyses of the effect using different EEG referencing methods (see Dimigen's studies), or by including a control condition where microsaccades occur without attention deployment (perhaps data recorded during ISIs could be used?).

3. Finally, I believe that the conclusion that the N2pc is not caused by microsaccades should be stated more cautiously. The N2pc was abolished when trials with microsaccades in the 150-400 ms window were excluded. The absence of an N2pc cannot be easily attributed to a low number of trials, as no-microsaccade trials comprised nearly 40% of the dataset—similar to the microsaccade conditions. When a shorter 200-300 ms window was examined, the N2pc was only marginally significant, making it possible that residual activity or small undetected microsaccades contributed to the effect. Finally, if Lambda waves were involved, their initial effect would be expected around 100 ms after an eye movement, which could influence the interpretation. A more cautious interpretation acknowledging these uncertainties would make the conclusions more robust. 

Minor comments: 

1. Citations - Olaf Dimigen is a pioneer in this field. He was the first to suggest that ERP components could be influenced by the Lambda wave. His 2009 Journal of Neuroscience paper should be cited and credited in the introduction.

2. A figure displaying the number of trials per time window for different microsaccade conditions would help clarify the feasibility of comparisons. Alternatively, a random sampling approach could be used to equalize trial numbers across conditions, though this may not be necessary given the already similar trial distributions.

3. The citation of our paper from 2009 (Yuval-Greenberg et al., 2009, Neuron) in reference 40 is inaccurate. That study did not establish a direct link between microsaccades and neural activity but instead demonstrated that oculomotor muscle contractions during microsaccades can appear as EEG signals and be mistakenly interpreted as brain activity. 

Best regards,

Shlomit Yuval-Greenberg

---

## [Decision Letter · Decision Letter 2]

20 Aug 2025

**If this is a Short Report, please check the number of figures and request the authors to reduce the number of figures down to 4 if they have more*** DELETE BEFORE SENDING

***IS THE AE ASSIGNED? IF NOT CANCEL THIS DECISION AND GO BACK!***

***EDIT AS REQUIRED***

Dear Dr Liu,

***EITHER***

Thank you for your patience while your manuscript "Microsaccades strongly modulate but do not directly cause the N2pc marker of spatial attention" went through peer-review at PLOS Biology. Your manuscript has now been evaluated by the PLOS Biology editors, an Academic Editor with relevant expertise, and by several independent reviewers.

***OR (if revision)***

Thank you for your patience while we considered your revised manuscript "Microsaccades strongly modulate but do not directly cause the N2pc marker of spatial attention" for consideration as a Short Reports at PLOS Biology. Your revised study has now been evaluated by the PLOS Biology editors, the Academic Editor [and the original reviewers - EDIT AS APPLICABLE]. 

****

In light of the reviews, which you will find at the end of this email, we are pleased to offer you the opportunity to address the [comments/remaining points] from the reviewers in a revision that we anticipate should not take you very long. We will then assess your revised manuscript and your response to the reviewers' comments with our Academic Editor aiming to avoid further rounds of peer-review, although we might need to consult with the reviewers, depending on the nature of the revisions.

**IMPORTANT - SUBMITTING YOUR REVISION**

*Resubmission Checklist*

*Published Peer Review*

*PLOS Data Policy*

*Blot and Gel Data Policy*

Sincerely,

Christian

Christian Schnell, PhD, 

Senior Editor

PLOS Biology

cschnell@plos.org

REVIEWS:

Reviewer's Responses to Questions

PLOS authors have the option to publish the peer review history of their article (what does this mean? ). If published, this will include your full peer review and any attached files.

**Do you want your identity to be public for this peer review?** For information about this choice, including consent withdrawal, please see our Privacy Policy .

Reviewer #1: Yes: Leon Y Deouell

Reviewer #2: No

Reviewer #3: Yes: Shlomit Yuval-Greenberg

Reviewer #1: The revised version of the manuscript went a long way in answering my major concerns. The addition of the sorted trials analysis ("ERP image") illuminates the fact that the lateralized activity (the N2pc) cannot be a simple artifact of the microsaccade, as the N2pc in many cases precedes the saccade onset. If I were the authors, I would emphasize this point. That is, not only that the N2pc does not show up concomitantly or after the saccade, but in some cases it may occur even before. This, together with the added details about microsaccades detection, strengthens the claims of the paper.

That said, the picture will be incomplete without the same ERP image of the "saccades away" trials and "no saccades" trials. In the towards image (right panel of figure 4), there is a clear lateralization (blue) time-locked to the saccade onset with a delay of about 200 ms (see the right side of the right panel in figure 4) - as if each towards saccade elicits an N2pc of its own! What happens in the away saccades? I am not asking just because of curiosity, but because there is still a conundrum remaining - the argument is that a) the N2pc is a genuine signature of attention (and not of eye movements), and b) that attention was equally and adequately shifted in the direction of the cue in the trials with away saccades. These two together predict an N2pc in the away saccade trials, but no evidence of this is found. In the final part of the discussion the authors address this by suggesting some form of 'masking' by summation of potentials on the scalp, but without a more detailed explanation of such masking, it remains mere hand waving. The authors could for example look at (micro)saccade-related-potentials (SRPs) from before the cue, and see if the lateralization of towards vs away saccades (relative to the ensuing cue, but before attention was shifted), can explain why the N2pc seems enhanced with towards saccades and diminished with away saccade. 

Also, showing the EOG image for the no saccades and away saccades will be illustrative in showing that the saccade detection algorithm is consistent with what the EOG shows. So basically, together with my previous comment, what I suggest is figure 4 to be expanded to show all 3 trial bins (separately). 

An additional comment for the authors to consider (although I don't insist) is that when they say that the N2pc is 'modulated' by microsaccades, it sounds contradictory to the claim that the microsaccades do not cause\directly elicit the N2pc. An alternative could be to say that the N2pc amplitude covaries with presence\direction of microsaccades. 

Some minor comments:

Whenever you mention eye muscle artifacts, I think you should also mention the possibility of a corneo-retinal dipole movement (which is also apparently ruled out). 

In line 308 when you say 'This resonates with recent studies" the reader could be helped by a few words on what these studies found. This is an important context for your study.

Line 163 - 'consisted' should be 'consistent'

Line 392 - ref 33 is noted twice. "(cf. [33, 33])".

Lines 387 - 394 - If I haven't read the comments and your resposnes to reviewer 2, I would not have understood the argument here. This needs to be clearer 

Reviewer #2: The authors have largely answered my questions.

The stuff about N2PC alignment to the stimulus (Fig. 4) is not really surprising/convincing to me. By definition, the n2pc is a stimulus-locked phenomenon, so why would it not remain locked to the stimulus onset in this new analysis? i.e. there are cases where saccades strongly influence visual sensitivity in such a way that the visual response remains clearly time-locked to stimulus onset but still strongly modulated by the saccades. This is, for example, how it happens with saccadic suppression. The stimulus onset occurs peri-saccadically, but the (strongly) modulated visual response to the stimulus remains clearly stimulus-locked. Anyway, I just felt that this needs to be clarified.

By the way, reference 31 is completely wrong and much less relevant to the current study! The actual correct reference should be: Zhang, T., Tian, X., Malevich, T., Baumann, M. P., & Hafed, Z. M. (2024). Foveal action for the control of extrafoveal vision. BioRxiv. I certainly hope that the authors read it, because it's much more relevant to the current study, and also especially because the authors wrote in their response letter that they actually read it.

Reviewer #3: The authors have addressed all of my concerns, and I find the manuscript ready for publication. Congratulations on an excellent study!

---

## [Editor Report · Decision Letter 3]

4 Sep 2025

Dear Baiwei,

Thank you for your patience while we considered your revised manuscript "Microsaccades strongly modulate but do not directly cause the N2pc marker of spatial attention" for publication as a Short Reports at PLOS Biology. This revised version of your manuscript has been evaluated by the PLOS Biology editors and the Academic Editor.

Based on our Academic Editor's assessment of your revision, we are likely to accept this manuscript for publication, provided you satisfactorily address the following data and other policy-related requests:

* We would like to suggest a different title to improve its accessibility for our broad audience: 

Microsaccades strongly modulate but do not necessarily cause the EEG N2pc marker of spatial attention

* Please add the links to the funding agencies in the Financial Disclosure statement in the manuscript details.

* Please include the approval/license number of the ethical approval from the institutional review board.

* Please include information in the Methods section whether the study has been conducted according to the principles expressed in the Declaration of Helsinki.

* DATA POLICY:

Regardless of the method selected, please ensure that you provide the individual numerical values that underlie the summary data displayed in the following figure panels as they are essential for readers to assess your analysis and to reproduce it: 2CG, S1 and S2.

* CODE POLICY

* Please note that per journal policy, the model system/species studied should be clearly stated in the abstract of your manuscript. 

We expect to receive your revised manuscript within two weeks. 

*Published Peer Review History*

*Press*

Sincerely,

Christian

Christian Schnell, PhD

Senior Editor

cschnell@plos.org

PLOS Biology

---

## [Editor Report · Decision Letter 4]

15 Sep 2025

Dear Baiwei,

Thank you for the submission of your revised Short Reports "Microsaccades strongly modulate but do not directly cause the EEG N2pc marker of spatial attention" for publication in PLOS Biology. On behalf of my colleagues and the Academic Editor, Ziad Hafed, I am pleased to say that we can in principle accept your manuscript for publication, provided you address any remaining formatting and reporting issues. These will be detailed in an email you should receive within 2-3 business days from our colleagues in the journal operations team; no action is required from you until then. Please note that we will not be able to formally accept your manuscript and schedule it for publication until you have completed any requested changes.

PRESS

Sincerely, 

Christian

Christian Schnell, PhD

Senior Editor

PLOS Biology

cschnell@plos.org